# Modification Effects of Carbon Nanotube Dispersion on the Mechanical Properties, Pore Structure, and Microstructure of Cement Mortar

**DOI:** 10.3390/ma13051101

**Published:** 2020-03-02

**Authors:** Shaowei Hu, Yaoqun Xu, Juan Wang, Peng Zhang, Jinjun Guo

**Affiliations:** 1School of Water Conservancy Engineering, Zhengzhou University, Zhengzhou 450001, China; hushaowei@cqu.edu.cn (S.H.); yaoqunxu_zzu@163.com (Y.X.); zhangpeng@zzu.edu.cn (P.Z.); guojinjun@zzu.edu.cn (J.G.); 2College of Civil Engineering, Chongqing University, Chongqing 400045, China

**Keywords:** carbon nanotubes, cement, mechanical properties, pore structure, microstructure

## Abstract

Carbon nanotubes (CNTs) are very effective in improving the performance of cement-based materials. Mechanical properties and pore structure were investigated for cement mortar with CNTs. Meanwhile, the composite morphology of CNT–cement material and the evolution of hydration products were observed by scanning electron microscope (SEM), and the quantitative relationship between mechanical properties and pore structure was analyzed. The results indicated that the strength of mortar increased with the addition of 0.05% CNTs and decreased when the fraction of CNTs increased to 0.5%. The porosity of mortar with dispersed CNTs increased significantly, as these pores may be introduced by the dispersant. The quantitative relationship between porosity and strength proved that the increased porosity is the reason for the decreased strength of mortar with 0.5% CNT content, while mortar matrix strength with 0.05% and 0.5% CNTs increased by 44.03% and 71.18%, respectively. SEM images show that CNTs are dispersed uniformly in the mortar without obvious agglomeration and that the CNTs and hydration products form a meshwork structure, which is the mechanism by which CNTs can enhance the strength of the cement matrix.

## 1. Introduction

Carbon nanotubes (CNTs) were discovered by Japanese scholar Lijima in 1991 [1]. CNTs are fibrous one-dimensional nanomaterials, with a diameter of 20–100 nm and a very high aspect ratio (100–1000). CNTs have excellent mechanical properties. Their tensile strength can reach 50–200 GPa, which is a factor of 100 greater than that of steel, and their density is only 1/6 that of steel, with an elastic modulus of >1 TPa [2]. The performance of CNTs is better than that of the most other fibers commonly used in research on reinforced composites. 

In engineering applications, researchers often use fiber materials to enhance the strength and toughness of cement-based composites [3,4], but macrofibers cannot fundamentally change the microdefects and cracks inside the cement. CNTs are a promising candidate for modified cement materials because of their excellent mechanical properties [5]. Many investigations have shown that CNTs can improve the mechanical performance [6,7] and durability [8,9,10,11] of cement-based materials. CNTs have excellent fiber material properties, effectively improving both toughness [12] and crack resistance of cement materials [13]. The amount of CNTs used by scholars in the cement is usually 0.01%–0.5% in terms of mass ratio [14,15], offering an increase in strength of 15%–50% [5,6,16]. At present, the optimal amounts of CNTs for cement composite are not consistent, and there is no consistent conclusion about the effect of CNT content on strength. The effect of CNTs on the microstructure and pore structure of cement-based materials has attracted considerable research attention. Through scanning electron microscopy (SEM) observation of a CNT–cement-based material matrix, it was observed that CNTs bridged in the cracks, preventing crack growth and increasing material toughness [17]. CNTs are covered and cemented by cement hydration products in cement-based materials, thus demonstrating the effective mechanical properties of CNTs [18]. CNTs are nanometer in size, enabling them to effectively fill microscopic pores in cement-based materials [15]. Thanongsak’s results indicate that the total porosity of CNT–cement composites decreases with increasing CNT content [19]. Xu analyzed CNT–cement material mercury intrusion porosimetry (MIP) test results and found that increasing the ratio of gel pore porosity to total porosity could improve its mechanical properties [14]. Other scholars have also observed dispersion of CNTs in cement-based materials and in hydration products of cement pastes [14,20].

Nanomaterials have nanoscale (1–100 nm) size, high atomic numbers, and interface energy on the surface of materials [21,22], therefore the interaction between monomers is strong, facilitating aggregation, which is also one of the key reasons for the unsatisfactory application of nanomaterials in cement-based materials [23]. As a result of the van der Waals force, CNT monomers become easily intertwined and agglomerated [24]. In Ibarra’ study, undispersed CNTs reduced the strength [25]. Shao prepared a CNT aqueous dispersion using ultrasound and a dispersant and mixed the CNT dispersion into cement-based materials, enhancing CNT dispersion [26]. Dispersants introduce air bubbles in cement-based materials, which leads to increased porosity and is disadvantageous to the development of material strength. In addition, a few scholars have focused on the effect of CNT dispersants on the cement matrix and have found the properties of cement are closely related to the pore structure [27,28,29]. 

In this work, the influence of CNT dispersion on cement mortar was studied. Mechanical properties were investigated, and pore structure measurements were made. SEM analysis at the microscale enabled the composite morphology of the CNT–cement material and the evolution of hydration products to be identified. The quantitative relationship between mechanical properties and pore structure was then analyzed. This work further confirms that CNT modification effectively improves the mechanical properties of cement-based materials, that CNT dispersion increases porosity, and that the influence of dispersants should considered.

## 2. Experimental 

### 2.1. Raw Materials

To achieve good dispersion in the mixture, CNTs were used with dispersants (Times Nano water dispersant; TNWDIS), in which the weight of dispersant was 0.2 times that of CNTs. The properties and Transmission Electron Microscope (TEM) images of CNTs dispersion (Chengdu Organic Chemicals Co., Ltd., Chengdu, China) are shown in Table 1 and Figure 1. The purity and outer diameter of CNTs was 98% and 30–80 nm respectively. Reference cement P.I 42.5 (by Chinese standards [30], Shandong Lucheng Cement Co., Ltd. Shandong, China) with 80 μm of mass ratio and 0.3% sieve residue was used, the main properties of which are shown in Table 2. Compared with cement, the particles size of CNTs are extremely small, therefore it is suitable for filling the micropores of cement matrix. Natural river sand (Tanghexian Xinmiao Shashi Co., Ltd, Henan, China) was used as fine aggregate, the main properties of which are given in Table 3. A high-range water reducing agent (Sobute New Materials Co., Ltd, Jiangsu, China) was used to adjust the workability of fresh cementitious composites.

### 2.2. Sample Preparation

For nanomodification, the CNTs were 0.05% and 0.5% of cement by mass (C1 and C2), and ordinary mortar (C0) was used for comparison. Three groups were prepared with the same water/cement ratio (w/c) of 0.5. Each group used 5400 g of sand and 1800 g of cement. The water consumption of each group of test samples was controlled to be 900 g. Table 4 lists the mix proportions of each mortar group. The sample molding was performed with the standard “Method of testing cements—Determination of strength” (Chinese national standard GB/T 17671-1999) [31].

### 2.3. Testing Procedures

#### 2.3.1. Mechanical Strength Test

The specimens’ size was 40 × 40 × 160 mm for compressive and flexural strength testing, and three samples of each group were tested. For each group, 12 mortar samples were casted; these were tested at 3, 7, 14, and 28 days to determine the strength development curves under compressive loads and flexural loads respectively. The WHY-300 (Shanghai Hualong Test Instruments Co., Ltd, Shanghai, China) microcomputer-controlled electro–hydraulic servo universal tester with error of ±1% was used to load the specimens. The loading rate was 5 × 10^−6^ s^−1^. Compressive and flexural failure samples are shown in Figure 2.

#### 2.3.2. Pore Structure Measurement

The linear traverse method (LTM) was employed to measure the pore structure of the hardened mortar. Mortar samples with a curing period of >28 days were taken for pore structure measurement. Two slices of 15–20 mm in thickness were cut from each broken specimen of the bending test. Slices at the center and edge of the mortar sample were used for observation. The area taken for the pore structure measurement was 30 × 30 mm at the center of the mortar slice. The location of the slices and the area of measurement are shown in Figure 3.

The collected mortar slices were ground and polished with a lapping machine. The rotary speed was set to 50 rpm, and the grinding and polishing times were 30 min each. We dried and blackened the surface after grinding and polishing by setting the temperature to 50 °C and drying for 3 h. An appropriate amount of white barium sulfate powder was sprinkled on the measurement surface of the slice and excess powder on the surface was removed after pressing. The slice was put into a pore structure automatic analyzer fixture to observe and calculate the pore size distribution and porosity in the measurement area. The LTM was implemented in accordance with the “Test code for hydraulic concrete” (Chinese national standard SL352-2006) [32].

#### 2.3.3. SEM Test

The microstructure of the mortar samples was observed by SEM. The samples used for the SEM test were also selected at four ages of 3, 7, 14, and 28 days to observe the effect of CNT dispersion on the evolution of cement hydration products. When each sample reached its corresponding period, it was taken out and broken into microscopic particles with a volume of <1 cm^3^, and the sample was stored in ethanol to prevent hydration. The sample preservation method is shown in Figure 4a. The sample was dried under vacuum at 50 °C and observed with SEM scanning. Representative sample fragments for SEM are shown in Figure 4b; it can be seen that the colors of C0, C1, and C2 deepen with the increase of CNT content.

## 3. Results

### 3.1. Mechanical Properties

The compressive strength test results for C0, C1, and C2 at stages of 3, 7, 14, and 28 days are shown in Figure 5. The compressive strength increased at 0.05% CNT content. Relative to C0, the compressive strength was enhanced by 16.3% at 3 days, 14.3% at 7 days, 10.1% at 14 days, and 2.4% at 28 days. It was obviously that the compressive strength of the cement mortar was enhanced by 0.05% CNT content addition, and, as the age increased, the enhancement effect weakened. At 0.5% CNT content, the compressive strength significantly decreased. Relative to C0, the compressive strength was reduced by 63% at 3 days, 53.1% at 7 days, 51.6% at 14 days, and 43.7% at 28 days, respectively. Clearly, 0.5% CNT content was detrimental to the compressive strength of the mortar.

The flexural strength test results for C0, C1, and C2 at stages of 3, 7, 14, and 28 days are shown in Figure 6. At 0.05% CNT content, the flexural strength increased. Relative to C0, the flexural strength was enhanced by 20.7% at 3 days, 19.3% at 7 days, 17.5% at 14 days, and 9.6% at 28 days, respectively. Clearly, at 3, 7, and 14 days, the flexural strength of the cement mortar was improved significantly by 0.05% CNT content addition, but, at 28 days, there was only a relatively small increase in flexural strength. Consistent with compressive strength, at 0.5% CNT content, flexural strength also significantly decreased. Relative to C0, the flexural strength was reduced by 65.9% at 3 days, 35.2% at 7 days, 31.8% at 14 days, and 28.5% at 28 days. Clearly, the addition of 0.5% CNTs was also detrimental to flexural strength of the mortar, but, as the age increased, the negative impact on C2 tended to weaken. 

The results of the mechanical properties tests demonstrate that the strength of the C1 mortar was improved, and the improvement in flexural strength was more significant. CNTs have excellent mechanical properties; as a reinforcing microfiber of cement-based materials, microfibers inhibit microcrack propagation in cement-based materials and form a meshwork microstructure to enhance the mortar matrix, consistent with literature results [14,19]. With a decrease of particle size, the atoms ratio on the particle surface and the reactivity of nanomaterial increase [33]. However, the mechanical properties of C2 mortar greatly reduce; thus, in this study we focus on this phenomenon. The significant increase in the porosity of the C2 mortar can be observed, which is considered to be the key to the lower C2 strength. 

### 3.2. LTM Results

The pore distributions of samples C0, C1, and C2 are shown as Figure 7, in which the black is the mortar matrix and white is the pores. It can be intuitively indicated that the porosity of C1 and C2 is significantly higher than that of C0, and, as the amount of CNTs increased, the porosity increased. The pores of C1 were relatively small and scattered, whereas those of C2 were dense and interconnected. 

The total porosity of hardened mortars is shown as Figure 8. Comparing the porosity of the center slice and the edge slice, one sees that there is not much difference, with center slice porosity being slightly higher than that of the edge slices. The porosities of center and edge slices of C0 were 6.69% and 6.22%, respectively, whereas those of C1 were 15.5% and 14%, respectively, and those of C2 were 43.26% and 38%, respectively. The porosity of 0.05% CNT mortar was more than twice that of ordinary mortar, and the porosity of 0.5% CNT mortar was a factor of 5–7 greater than that of ordinary mortar.

The pore size distributions represented by pore radius distribution, and relative to porosity in the LTM test, are shown as Figure 9. It can be seen that C1 and C2 incorporating CNTs had higher porosity and higher peak values compared with those of C0. It can also be noted that C2 had a higher porosity than that of C1. The most probable cause for this is that pore size increased gradually with the increase of CNT content, with the curves shifting rightward to reflect the roughening of the pores. This is consistent with Figure 7 and Figure 8. It is suggested that the dispersant TNWDIS leads to increased porosity and the dispersant has a strong air-entraining effect [14,26]. 

The measured pores were divided into six sizes based on their radius: <50, 50–100, 100–200, 200–500, 500–1000, and >1000 µm. The relative porosity distributions according to these size ranges are shown in Figure 10. The relative porosity of macropores (with pore sizes of 500–1000 and >1000 µm) increased with the incorporation of CNTs. Compared with C0, the proportions of pores (>1000 µm) of the samples with the incorporation of CNTs range from 0%–3% (C0) to 5%–10% (C1) and 15%–24% (C2), while the micropores of C0, C1, and C2 account for 11%–14%, 16%–21%, and 7%–10%, respectively. The results indicate that the pores introduced by the CNT dispersion mainly consist of mesopores of 100–500 µm and macropores of >500 µm. 

The porosity and pore size distribution have an important influence on the strength of cement-based materials [27]. The trends of porosity, compressive strength and flexural strength at 28 days are shown as Figure 11. It can be seen that with the increase of CNTs content, the porosity increased, while the compressive and flexural strengths increased first and then decreased. When the mortar was loaded, the effective loading area and mortar strength had a negative correlation with porosity. Therefore it was necessary to further analyze the quantitative relationship between strength and porosity in the next section.

### 3.3. SEM Results

#### 3.3.1. CNTs in Cement-Based Materials

SEM observations of CNTs in C1 and C2 mortars are shown in Figure 12. It can be seen that CNTs are cemented with the hydration products calcium silicate hydrates (C–S–H) gel and calcium hydroxide (CH), and CNTs cement with each other in the matrix to form a meshwork microstructure. Other research results [13,14,34] suggest that this meshwork microstructure can absorb energy when the cement-based materials are under stress, as pulling work of CNTs in the matrix when a crack develops. This indicates that CNT microfibers enhance the toughness of materials, which agrees with the results of the significantly improved flexural strength of C1 in this study.

By comparing the microstructure of CNT–cement composites in C1 and C2, it can be intuitively seen that the quantity of CNTs in the microstructure of C2 is higher than that of C1. The trend in the density of CNTs in composites is consistent with that of CNT content; this also implies the efficient dispersion of CNTs in the mortar matrix, with no aggregates such as CNT clusters or clumps being seen in the SEM images.

#### 3.3.2. Morphology of Hydration Products

The morphology of ordinary mortar C0 and CNT-modified mortars C1 and C2 at the four phases of 3, 7, 14, and 28 days are shown in Figure 13. In this study, the micro-morphological evolution process of cement hydration products was observed using a lower magnification.

In the early hydration period of C0, the structure of hydration products was loose; at 7 days, numerous tiny needle–rod ettringite and hexagonal plate calcium hydroxide crystals could be observed. With the development of hydration, the hydration product C–S–H gradually increased at 14 and 28 days, and the microstructure tended to be dense, which conforms to the description of the cement hydration process in the literature [35]. It can be seen in C1 that, in the early stage of hydration, at ages of 3 and 7 days, the hydration product was loose, and the microstructure was poor in density. With the development of the hydration process, at ages of 14 and 28 days, the content of C–S–H increased, and the denseness of the microstructure was significantly improved, which is consistent with the development process of the hydration products of the ordinary mortar C0. In the C2 images, during early hydration, the hydration products consisted of a large number of CH molecules and the hydration products were loose, with poor compactness of the microstructure. At 28 days, as shown in Figure 13, the compactness of hydration products increased.

## 4. Discussion

The results above show that an appropriate amount of CNT dispersion can effectively improve the mechanical properties of the mortar, with the introduced pores increasing the porosity of the mortar. However, the significant increase in porosity caused by excessive use of CNT dispersions will significantly reduce the strength of mortar. Based on the above test results, a quantitative relationship between strength and porosity of CNT-modified mortar is now discussed.

### 4.1. Quantitative Relationship between Strength and Porosity

High porosity can adversely affect the mechanical properties of building materials, and many equations have been established to quantify the relationship between strength and pore structure [36,37,38]. Based on the mechanical properties and LTM test results of CNT-modified mortar, the relationship between strength and porosity of CNT-modified mortar was analyzed. The relationship between proposed by Tang [37] and Jin [39] was used. 

In this study, the LTM was used in pore structure measurements of CNT-modified mortar, and mortar was considered a two-phase composite composed of mortar and pores. The spherical pore model was used to describe the pores tested by the LTM. The mortar was considered to be consisting of *n* porous bodies, with pores of the same pore size distributed into the same porous body, and the porous bodies were connected in parallel. The radius *r* is arranged from large to small, and *B_n_* refers to the *n*th porous body, as shown in Figure 14.

For the above model, the effective loading area ratio *A_m_* of the mortar matrix can be expressed as Equation (1).
(1)$$A_{m}=1-\pi^{1/3}{\left(\frac{3}{4}v\right)}^{2/3}$$
where *v* is the porosity, and the relationship between the mortar elastic modulus *E* and the matrix elastic modulus *E_m_* can be expressed as Equation (2).
(2)$$E=E_{m}A_{m}$$

It will fracture when the stress is greater than the critical stress given by Equation (3), which can be simplified to Equation (4).
(3)$$\sigma_{c}=\sqrt{\frac{{2E}_{m}A_{m}\gamma }{\pi kr}}$$
(4)$$\sigma_{c}=\sqrt{{KA}_{m}/r}$$
where *K* is a constant dependent on the matrix elastic modulus *E_m_*, surface energy *γ*, and the pore shape constant *k*. For pores equivalent to spherical pores, *k* is a constant value. The value of *K* indicates the strength of the mortar matrix. In this study, *K* is the matrix strength index.

At the beginning of loading, the effective area of element *B*_1_ with a porous body radius *r*_1_ is the smallest. When *P*_1_ is applied on a porous material, the normal stress on the matrix can be calculated by Equation (5).
*σ* = *P*_1_/*A*_*m*1_(5)

If the normal stress of specimen satisfied the condition of Equation (6), element B1 would be destroyed and fail to bear loads, while the effective loading area ratio of the specimen would be reduced from *A*_*m*1_ to *A*_*m*2_ (Equation (7)). Steps are repeated until all the elements are destroyed and the compressive strength of the material can be drawn according to Equation (8).
(6)$$\sigma >\sqrt{{KA}_{m1}/r_{1}}$$
(7)$$A_{m2}=A_{m1}\left(1-\frac{V_{1}}{V}\right)$$
(8)$$\sigma ={P/A}_{mn}>\sqrt{{KA}_{mn}/r_{n}}$$

The fracture process in the porous body model agrees well with the results of the literature [37,39]. From the above analysis, it is difficult to ascertain the exact matrix strength index *K* theoretically; it can only be obtained by quantitative analysis based on experimental data.

### 4.2. Calculation and Result Analysis

For the calculation of strength and pore size distribution, consistent with the test results in Section 3.2, we measured the average pore size (*r_i_*), porosity (*V_i_*), total porosity (*V*), and compressive strength (*f_c_*; see Table 5). *V_i_* was the porosity of the given pores size, such as the *V_1_* in Equation (7). The quantitative relationship between pore structure and strength is very difficult to calculate manually, so a computer program was used to solve the problem. Based on the pore size distribution and porosity measured in the test and the compressive strength of the mortar, the matrix strength index *K* of the C0, C1, and C2 test groups was iteratively calculated.

The compressive strength can be calculated by applying Equations (4) and (8) with a given value of *K* and the parameters in Table 5. What needs to be stated is the value of *K* should be greater than 1 considering the value range of compressive strength of mortar. Moreover, the curves were obtained and shown as Figure 15. It can be found that the relationships between the matrix strength index *K* and the compressive strength at the center of the specimen are similar with that of the edges of the specimen. The results show that, as the matrix strength index *K* increases, the compressive strength increases. For the same *K*, the higher the porosity, the greater the compressive strength. The calculation results are in accordance with the basic understanding of the relationship between porosity and strength.

The matrix strength index *K* of C0, C1, and C2 obtained from the test results of compressive strength and porosity are shown in Figure 16. The relative matrix strengths of C0, C1, and C2 were 3.355–3.845, 5.165–5.205, and 6.22–6.105, respectively. At 0.05% and 0.5% CNT content, the matrix strength index *K* was enhanced by 44.03% and 71.18%, respectively. It can be concluded that, with the increase of CNTs, the effect of CNTs on the strength of the cement-based matrix becomes more significant.

The use of CNT dispersions increases the porosity of the mortar, but CNTs are not considered air-entraining, therefore the dispersants used in CNT dispersions should be chosen carefully. The CNT dispersant used in this study was TNWDIS, with a content that was 0.2 times the weight of CNTs, which means that 0.01% and 0.1% TNWDIS dispersants were contained in C1 and C2, respectively. The mechanism of this surfactant is that it can adsorb on the surface of CNTs and reduce the surface energy of CNTs, thereby promoting the dispersion of CNTs [40]. Moreover, it can form a directional arrangement at the two-phase interface, then form micelles and vesicles. This is also the mechanism of the air-entraining agent used in concrete, and it is also the action mechanism of superplasticizers in concrete [12,41]. However, the TNWDIS dispersant in the mortar has a strong air-entraining effect on cement-based materials, which has a serious adverse effect on the strength of the mortar.

## 5. Conclusions

The mechanical properties, pore structure, and microstructure of cement mortar reinforced with CNT dispersions were investigated and the quantitative relationship between mortar strength and pore structure was analyzed. The main results and conclusions are as follows:
(1)The mechanical properties of 0.05% and 0.5% CNT mortars were evaluated. The experimental results show that the strength of mortar is improved by adding 0.05% CNT, while the negative impact occurred with the addition content of CNTs up to 0.5%.(2)The pore structure of mortars was studied. The total porosity of mortar containing 0.05% ~ 0.5% CNTs were increased by 15% ~ 40% compared to that of the reference normal mortar. In addition, the increase of pores mainly appeared in the range of pores with diameter of 500–1000 um.(3)The microstructure of CNT–cement composites was observed. The CNTs can be well dispersed in mortar when they are used with dispersant, and CNT meshwork will form in cement matrix, the compressive strength of which can be improved accordingly.(4)There is no doubt that both of the strength of cement matrix and porosity of mortar are increase with the addition of CNTs. But the compressive strength of mortar is irregular under the combined effect of the matrix strength and mortar porosity. The strength decrease of the mortar with 0.5% CNTs is mainly due to the sharp increase of porosity, which may be caused by the use of dispersant. Therefore, it is necessary to study the application method of CNTs to take the advantage of the excellent improvement capability of CNTs to cement matrix strength.

## Figures and Tables

**Figure 1 materials-13-01101-f001:**
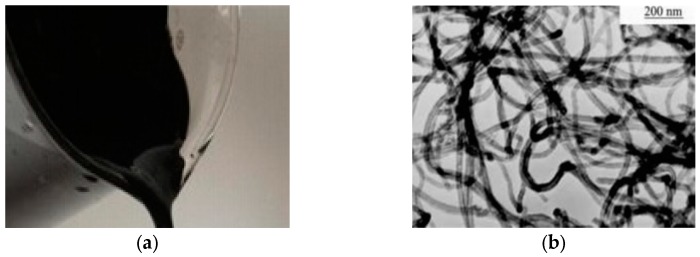
Carbon nanotubes (CNT) dispersion and Transmission Electron Microscope (TEM) image of CNTs. (**a**) CNT dispersion and (**b**) TEM image of CNTs.

**Figure 2 materials-13-01101-f002:**
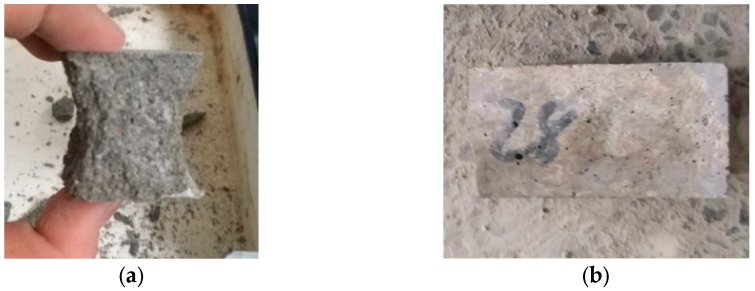
Compressive and flexural failure specimens. (**a**) Compressive failure specimen and (**b**) flexural failure specimen.

**Figure 3 materials-13-01101-f003:**
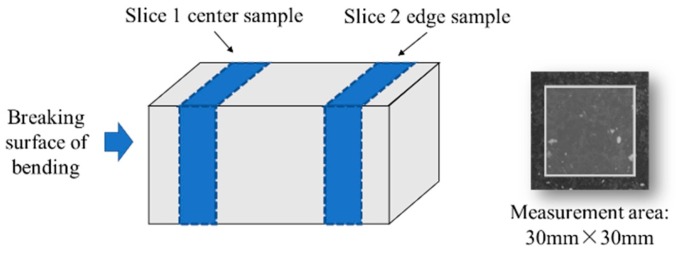
Location of the slices and the area of measurement.

**Figure 4 materials-13-01101-f004:**
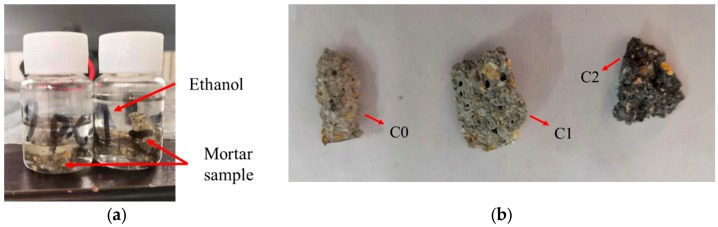
Scanning electron microscopy (SEM) micro samples. (**a**) Micro sample preservation method and (**b**) Sample fragments of C0 (with no CNTs), C1 (with 0.05% CNTs), and C2 (with 0.5% CNTs).

**Figure 5 materials-13-01101-f005:**
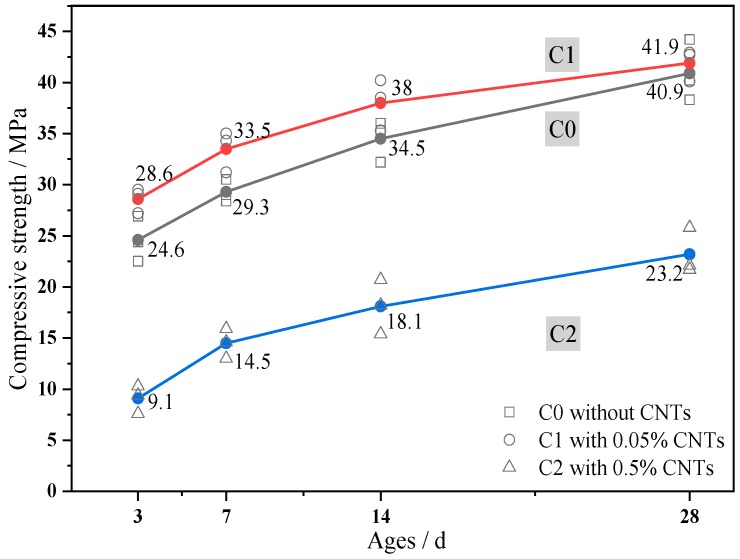
Development of compressive strength.

**Figure 6 materials-13-01101-f006:**
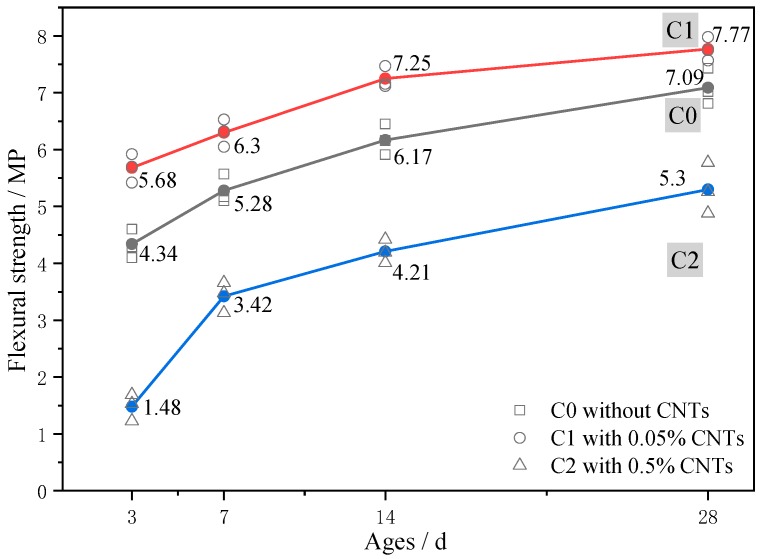
Development of flexural strength.

**Figure 7 materials-13-01101-f007:**
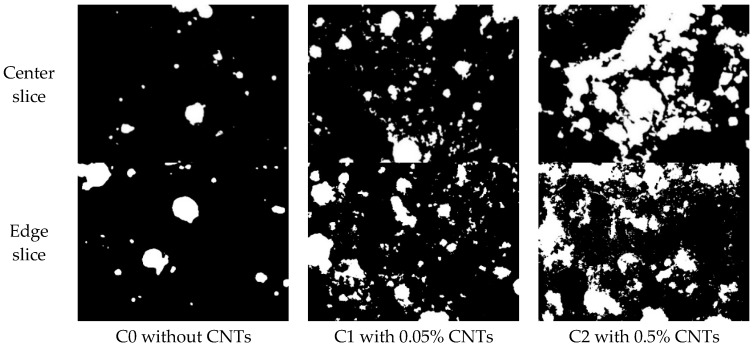
Binary images of sample pore distributions.

**Figure 8 materials-13-01101-f008:**
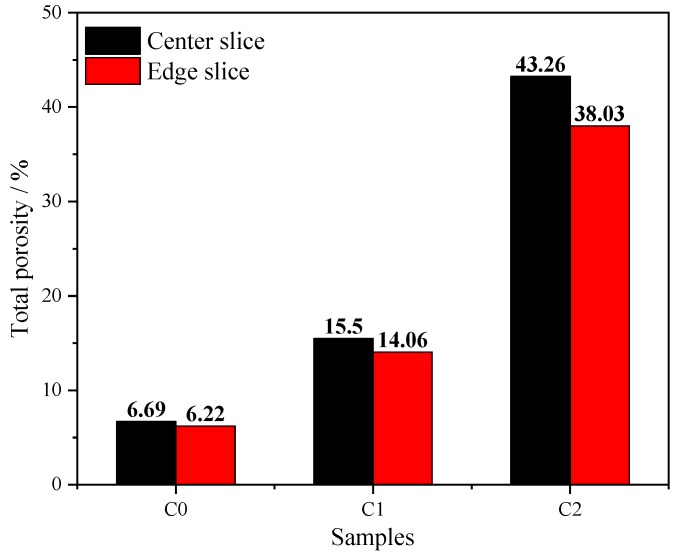
Total porosity of hardened mortar.

**Figure 9 materials-13-01101-f009:**
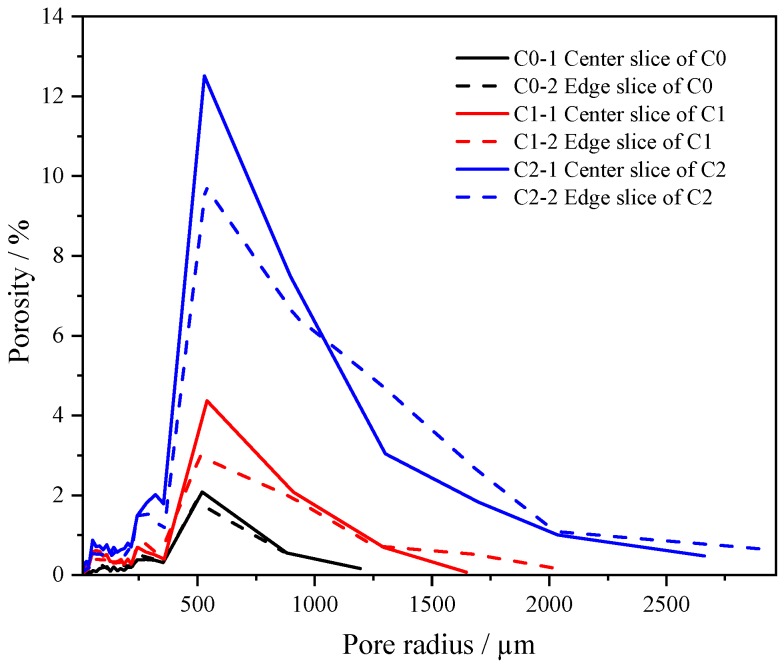
Porosity distribution in the linear traverse method (LTM) test.

**Figure 10 materials-13-01101-f010:**
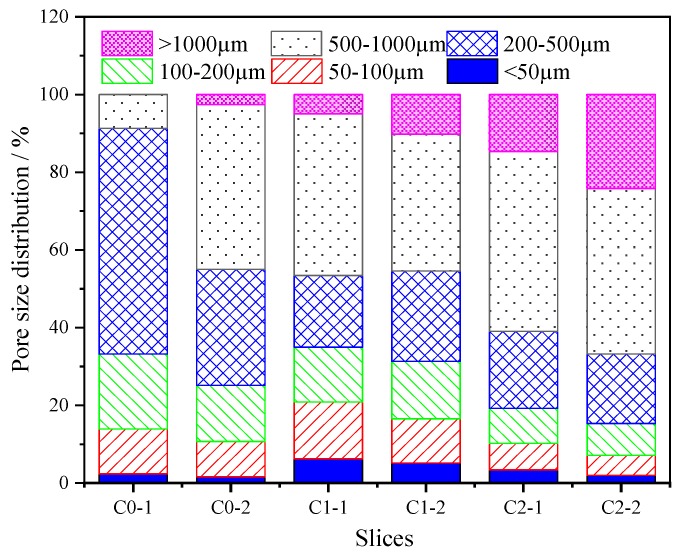
Pore size distribution of mortar.

**Figure 11 materials-13-01101-f011:**
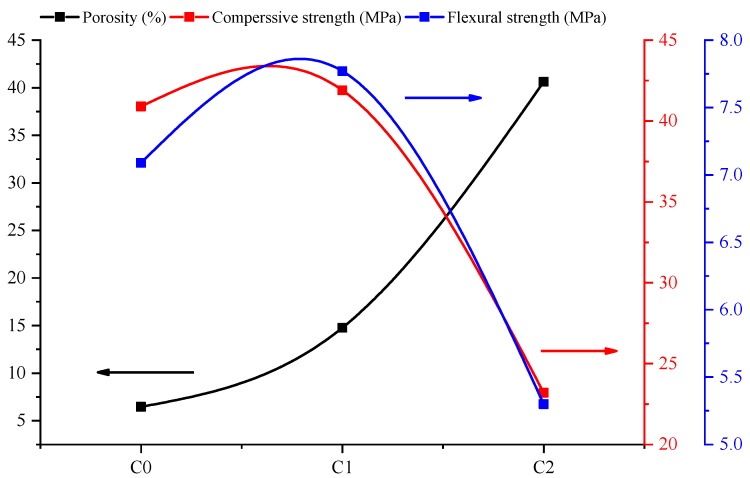
Trend curves of mortar porosity, compressive strength, and flexural strength: C0 without CNTs; C1 with 0.05% CNTs; C2 with 0.5% CNTs.

**Figure 12 materials-13-01101-f012:**
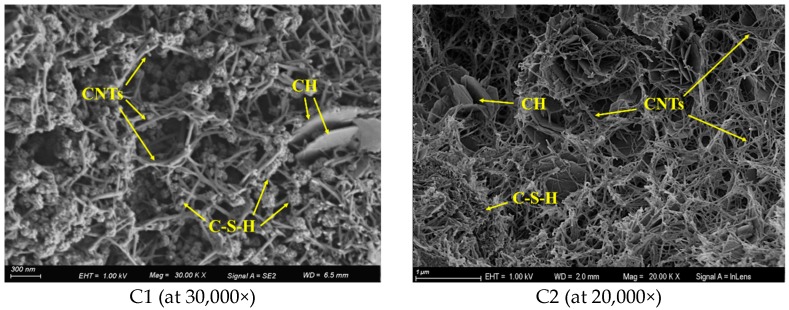
SEM images of CNTs in mortar.

**Figure 13 materials-13-01101-f013:**
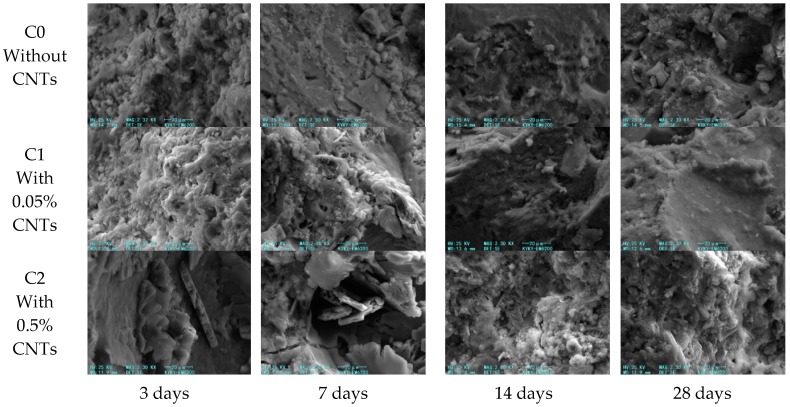
Morphology of hydration products in C0, C1, and C2 cement mortar.

**Figure 14 materials-13-01101-f014:**
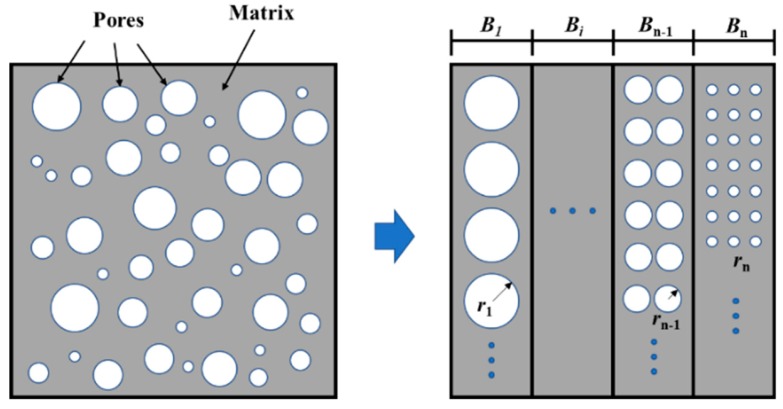
Two-phase spherical pore model and parallel porous body model.

**Figure 15 materials-13-01101-f015:**
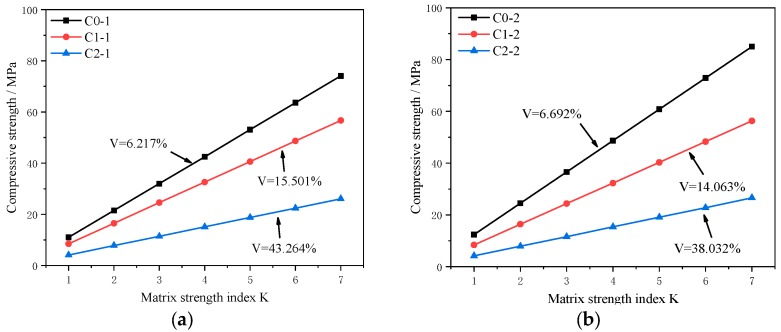
Relationship between *K* and the compressive strength. (**a**) Center slice and (**b**) edge slice.

**Figure 16 materials-13-01101-f016:**
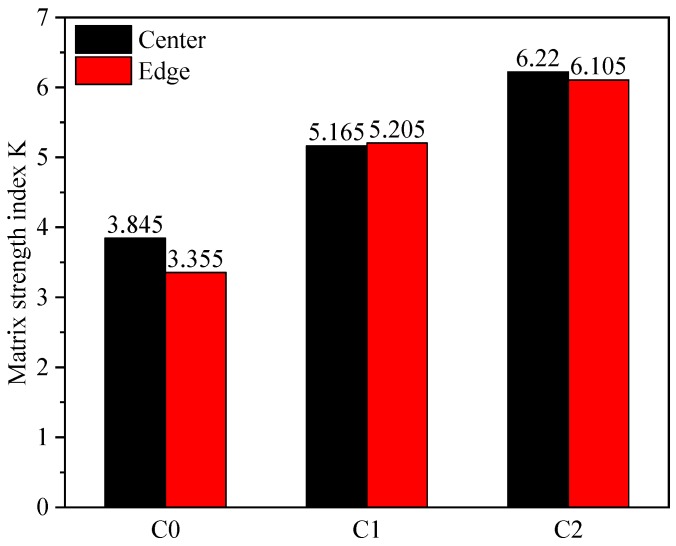
Matrix strength index *K*.

**Table 1 materials-13-01101-t001:** Properties of carbon nanotubes (CNT) dispersion.

Appearance	Outer Diameter (nm)	Purity	Length (µm)	Specific Surface Area (m^2^/g)	Tap Density (g/cm^3^)	True Density (g/cm^3^)	Solvent	Content
Black liquid	30–80	98%	10	>60	0.18	~2.1	Water	10%

**Table 2 materials-13-01101-t002:** Properties of cement.

Chemical Compositions	SiO_2_	Al_2_O_3_	Fe_2_O_3_	CaO	MgO	Na_2_O	SO_3_	Loss on Ignition	Specific Gravity	Specific Surface (cm^2^/g)	Bulk Density (g/cm^3^)
Composition (%)	20.56	4.6	3.23	62.56	2.57	0.59	2.95	2.94	3.13	3530	1.7

**Table 3 materials-13-01101-t003:** Properties of sand.

Aggregate Types	Visual Density (kg/m^3^)	Fineness Modulus	Water Absorption (%)	Mud Content (%)
Natural river sand	2744	2.94	0.58	3.7

**Table 4 materials-13-01101-t004:** Mix proportion of mortar for evaluating the effect of CNTs.

Specimen	Sand (g)	Cement (g)	Water/Cement	Admixture (g)	
				CNTs	Water Reducer
C0	5400	1800	0.5	0	18
C1	5400	1800	0.5	0.9	18
C2	5400	1800	0.5	9	18

Notes: C0 with no CNTs, C1 with 0.05% CNTs, C2 with 0.5% CNTs.

**Table 5 materials-13-01101-t005:** Parameters for quantitative calculations.

Size Range (μm)	C0-1		C0-2		C1-1		C1-2		C2-1		C2-2	
	*r_i_* (μm)	*V_i_* (%)	*r_i_* (μm)	*V_i_* (%)	*r_i_* (μm)	*V_i_* (%)	*r_i_* (μm)	*V_i_* (%)	*r_i_* (μm)	*V_i_* (%)	*r_i_* (μm)	*V_i_* (%)
<50	19.6	0.16	16.5	0.10	24.8	0.97	23.1	0.73	21.5	1.47	21.3	0.74
50–100	74.3	0.77	76.2	0.57	70.4	2.27	70.5	1.60	69.4	2.95	71.2	1.97
100–200	146.1	1.29	142.9	0.90	141.4	2.19	142.4	2.08	144.5	3.88	145.3	3.12
200–500	344.6	3.89	268.5	1.86	262.8	2.86	276.9	3.27	277.4	8.60	270.3	6.80
500–1000	866.4	0.58	569.4	2.64	621.7	6.45	615.2	4.95	624.5	20.01	642.4	16.22
>1000	None		1194.7	0.16	1321.4	0.77	1707.9	1.44	1.815	6.36	1844.5	9.18
*V* (%)	6.69		6.22		15.50		14.06		43.26		38.03	
*f_c_* (MPa)	40.9				41.9				23.2			

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
