# Peer review of "Modification Effects of Carbon Nanotube Dispersion on the Mechanical Properties, Pore Structure, and Microstructure of Cement Mortar"

_materials, 2020, doi:10.3390/ma13051101_

Round 1

Reviewer 1 Report

Ref. No.: materials-733745

Modification effects of carbon nanotube dispersion on the mechanical properties, pore structure, and microstructure of cement mortar

Reviewer comments:

SUMMARY

The manuscript deals with an investigation on the modification effects of carbon nanotube dispersion on the mechanical properties, pore structure, and microstructure of cement mortar. This is topic has not been widely covered in the literature. Then, it is a subject of great interest, but it is somehow limited in the analysis of these results.

MAIN IMPRESSIONS

From a scientific point of view, the following issues must be addressed: i) you should obtain relationships between the different parameters considered in the present study and discuss them with other studies, ii) The conclusion should be concise and engaging. Aim to leave the reader with a clear understanding of the main discovery. In addition, the conclusion should be where you describe how a previously identified gap in the literature. In this paper, some conclusions are repetitive.

MORE DETAILED COMMENTS

Line 73: Raw materials: Please provide the location (city, state, country) for all suppliers and manufacturers companies (cement, TNMDIS and so om). Please review and revise throughout the manuscript.

Lines 68 & 78: Please, define the TNMDIS acronym.

Line 77: In “CNTs were used as dispersants”, do you mean “CNTs were used with dispersants”?

Line 89: Add LOI in Table 2. (Loss on ignition).

Line 89: Specific gravity units are missing in Table 2.

Line 89: Please, add the size distribution of the cement and compare it with the CNTs size.

Lines 82&98&126 and so on: Could you please add the reference to the Chinese standards?

Line 149: Fig. 5 must be larger. It is not possible to read the captions.

Line 149: Fig. 5: The improvement of compressive strength at 28 days is extremely low (2.4%) with 0.05% CNT content. Do you think that this low improvement could be achieved with other cheaper and more available materials such as silica fume? Could you please discuss the effect of the particle size distribution in the compressive strength at 28 days? For instance, some authors have studied the effect of silica fume fineness on the improvement of Portland cement strength performance.

Line 161: Fig. 6 must be larger. It is not possible to read the captions.

Line 190: Comparing Fig. 5 & Fig. 8, there is not clear relationship between compressive strength at 28 days and porosity. Therefore, why do you justify the worse performance of 0.5% CNT by the higher porosity? C1 has a higher total porosity than C0 but also a higher compressive strength.

Line 190: Total pore size increased gradually with the increase of CNT content. However, pore size distribution shift rightward to reflect the roughening of the pores. I recommend presenting a graph showing the relationship between capillary pores (Fig. 10)  and compressive strength. This will help to demonstrate the actual effect of the carbon nanotube dispersion on the mechanical properties and pore structure.

Line 203: Fig 10 must be larger. It is not possible to read the captions.

Lines 305-338: Conclusions: Conclusions in lines 308 & 318 are the same. Conclusions in lines 309 & 319 are the same. The conclusion should be concise and engaging. Aim to leave the reader with a clear understanding of the main discovery. In addition, the conclusion should be where you describe how a previously identified gap in the literature. In this paper, some conclusions are repetitive.

Lines 317, 321, 326, 330 and 331 are not conclusions, they could be deleted.

RECOMMENDATION

In conclusion, Minor changes have been proposed.

Reviewer 2 Report

It is an interesting and well written paper aimed at the assessment of the modification effects of carbon nanotube dispersion on the mechanical properties, pore structure, and microstructure of cement mortar. The paper is generally well organized and the used methods were chosen properly for such type of research. Before the paper will be accepted for publication, following minor revisions might be conducted:

1) line 67 – please replace are studied with is studied

2) line 77 “CNTs were used as dispersants to achieve a uniform dispersion in the mixture” this statement is definitely wrong, CNTs were use as dispersed phase not as dispersant – for that purpose TNMDIS was used

3) What type of sand was used? What was it particle size, water absorption etc.? This information must be provided in revised manuscript.

4) line 103 – mortar cubes were cast? As introduced in line 102, prism specimens were used. Please, correct and explain.

5) line 103 – replace were cast with were casted

6) What was the measurement uncertainty of performed compressive and flexural strength tests?

7) The technical standard used for the mechanical tests might be introduced both in the text and list of references

8) What is the price of used CNTs? Is their contribution the improvement in mechanical strength efficient in respect to material cost? Please comment.

9) Information on the bulk density of the tested materials should be provided as usually required in testing the compressive strength of concrete.

10) Standard used should be introduced in the list of references.

11) It is not clear, how the strength inced K was calculated. Please explain.

12) Porosity Vi was porosity corresponding to the volume of pores of specific size range? Please, explain in text.
